# Comparison of the Inhibitory Activities of 5,6-Dihydroergosterol Glycoside α- and β-Anomers on Skin Inflammation

**DOI:** 10.3390/molecules24020371

**Published:** 2019-01-21

**Authors:** Tae Kyun Kim, Young Kyoung Cho, HoonGyu Park, Tae Hoon Lee, Hakwon Kim

**Affiliations:** Department of Applied Chemistry and Global Center for Pharmaceutical Ingredient Materials, Kyung Hee University, Yongin 17410, Korea; ocheon@yuhan.co.kr (T.K.K.); yk930504@naver.com (Y.K.C.); hungu860@naver.com (H.P.)

**Keywords:** chronic skin inflammation, chemokines, cytokines, dihydroergosterol, DHE-glycosides

## Abstract

Chronic skin inflammatory diseases, such as atopic dermatitis, are associated with a dysfunctional skin barrier due to an increase in various inflammatory stimuli, for instance inflammatory cytokines and chemokines. In particular, CCL17 and CCL22 expression is increased in patients with chronic skin inflammation. In this study, we synthesized several α- and β-anomers of dihydroergosterol (DHE)-glycosides and assessed their effects on CCL17 and CCL22 expression. We confirmed that the β-anomers of DHE-glycosides were superior to α-anomers of DHE-glycosides in inhibiting CCL17 and CCL22 mRNA and protein expression. In addition, we determined that DHE-glycoside β-anomers showed strong inhibitory activity towards pro-inflammatory cytokine mRNA and protein expression, including that of TNF-α, IL-6, and IL-1β- in stimulated HaCaT cells. These results imply that DHE-glycoside α- and β-anomers should be separated during synthesis of drugs for chronic skin inflammation. Our results also suggest that β-anomers of DHE-glycosides may play an important role as new drugs for chronic skin inflammation because of their ability to inhibit the skin inflammatory biomarker proteins CCL17 and CCL22.

## 1. Introduction

Natural products from various plants with soap-like properties, such as saponins, have played an important role in medicine and daily life for thousands of years [1]. Saponins are amphipathic glycosides grouped structurally by having one or more hydrophilic glycoside moieties combined with a lipophilic triterpene or steroid derivative, and have well characterized biological properties including hemolytic activity, molluscicidal activity, and anti-inflammatory activity [2].

We previously confirmed that spinasterol-glucose (**1**) (spinasterol-Glc, Figure 1), extracted and identified from natural herbs, not only exhibited potent anti-inflammatory activity but also inhibited CCL17 mRNA and protein expression in TNF-α/INF-γ-stimulated HaCaT cells [3]. However, spinasterol-Glc is difficult to obtain from either natural plant extracts or using synthetic methods. The synthesis of spinasterol-Glc is very difficult because its steroidal skeleton, α-spinasterol, is not readily available. Recently, synthesis of α-spinasterol from stigmasterol using a double bond migration method has been reported; however, the synthesis was still difficult and inefficient [4,5]. Therefore, we previously designed and synthesized a new sterol, 5,6-dihydroergosterol (DHE or stellarsterol) that is structurally similar to spinasterol and more easily obtained. In previous studies, we demonstrated that DHE and its glycosides have comparable anti-inflammatory activity to spinasterol-Glc. Of the synthesized DHE-glycosides, the β-anomer of DHE-Glc (5,6-dihydroergosterol-glucose (**2**) showed the greatest anti-inflammatory activity [6]. In addition, we confirmed that the expression of CCL 17 and CCL22 mRNA and protein were strongly inhibited in a DNCB-induced atopic dermatitis (AD) animal model [7].

Chronic skin inflammatory diseases are associated with a dysfunctional skin barrier that is characterized by elevated expression of various inflammatory stimuli, specifically pro-inflammatory cytokines and chemokines [8,9]. In skin conditions like chronic inflammation, small signaling proteins, including chemokines, are secreted. The main function of these chemokines is to recruit leukocytes, monocytes, and neutrophils to inflammatory sites [10,11]. Specifically, in abnormal skin conditions like AD, chemokines have been reported to promote the secretion of pro-inflammatory cytokines and thymic stromal lymphopoietin (TSLP) proteins that cause itching [12]. Thymus and activation-regulated chemokine (TARC/CCL17) are a type of C-C chemokine that uses CC chemokine receptor 4 (CCR4) as a receptor, and specifically binds to and induces chemotaxis in Th2 cells and is associated with Th2 cell-mediated inflammatory diseases, such as AD [13]. Another CC-chemokine, macrophage-derived chemokine (MDC/CCL22), shares homology (37% amino acid identity) with CCL17, and both chemokines mediate a variety of biological activities through CCR4 [14]. Previous reports on the various pharmacological and medicinal functions of CCL17 and CCL22 in skin inflammation suggest that these chemokines can be used as biomarkers of chronic skin inflammation [15,16,17]. Furthermore, CCL17 has been shown to be present at high levels in the blood of AD patients and highly expressed in skin lesions [15]. In addition, CCL22 expression was increased in the epidermis and serum of patients with chronic inflammatory skin diseases [16]. These results indicate that CCL17 and CCL22 play crucial roles in the mobilization of Th2 cells and that increased expression of these chemokines further exacerbates chronic skin inflammation in diseases like AD [18].

Generally, glycosides consist of glycone (sugar) and aglycone (non-sugar) components, and can be divided into alpha (α) and beta (β) anomers depending on the linking stereochemistry of the glycone and aglycone portions (Figure 1). In our synthesis process, we have always obtained both the α- and β-anomers of DHE-glycosides, particularly during the glycosylation step. Since there have been few reports describing the relative bioactivity data of α- and β-anomers of DHE-glycosides, we were interested in determining their relative biological activities. To do this, we compared the inhibitory activities of α- and β-DHE-glycosides on TNF-α/IFN-γ-induced expression of pro-inflammatory cytokines and chemokines, including CCL17 and CCL22, in TNF-α/IFN-γ-stimulated human keratinocytes. Here, we report the synthesis and anti-inflammatory activity of α- and β-anomers of DHE-glycosides, and which anomer of DHE-glycosides more effectively inhibits the gene expression of pro-inflammatory cytokines IL1-b, IL-6, and TNF-α, and the chemokines CCL17 and CCL22, in HaCaT cells. Our results indicate that β-anomers of DHE-glycosides inhibit the mRNA and protein expression of pro-inflammatory cytokines and chemokines in TNF-α/IFN-γ-activated HaCaT cells to a greater extent than the α-anomers of DHE-glycosides, and that this is likely because of the instability in solution of the α-anomers of DHE-glycosides. These findings may contribute to the development of useful therapeutic agents for chronic skin inflammation diseases such as atopic dermatitis.

## 2. Results

### 2.1. Synthesis of α- and β-Anomers of 5,6-Dihydroergosterol-Glycosides (DHE-Glycosides)

As shown in Scheme 1, β- and α-anomers (**2**–**5**) of DHE-glycosides can be prepared by acid-catalyzed Schmidt glycosylation of DHE with OH-protected sugars followed by deprotection.

Previously, we determined which of various glycosyl donors, including glycosyl halide, glycosyl sulfide, and glycosyl trichloroacetimidate, in the presence of several catalysts, resulted in the most efficient glycosylation of DHE. We found that when *O*-tetrabenzoyl-protected glycosyl trichloroacetimidates (**7** and **8**) were used as glycosyl donors and Cu(OTf)_2_ was used as a catalyst, an effective stereoselective glycosylation reaction occurred, especially for β-anomers [6]. Therefore, glycosyl trichloroacetimidates (**7** or **8**), glycosyl donors for Cu(OTf)_2_-catalyzed glycosylation, were used in this study. The two glycosyl donors, glucopyranosyl and galactopyranosyl trichloroacetimidate, and DHE were synthesized according to previous methods [6].

The anomeric effect in sugar chemistry predicts that α-anomers will be the major product of glycosylation reactions. However, β-anomers have been reported to be produced when acyl groups capable of neighboring group participation (NGP) are present at adjacent carbon positions. Unlike β-anomers, which are the major products in our reaction with a benzoyl group in an adjacent position, α-anomers are not readily available in sufficient quantities to carry out biological activity studies. According to the literature, α-anomers are primarily produced when a benzyl (Bn)-protected sugar is used as a glycosyl donor [19]. We attempted to glycosylate with a Bn-protected sugar, but there was an issue during the debenzylation step. Therefore, we examined the various reaction conditions under which α-anomers are obtained with a benzoyl (Bz)-protected sugar. Experimental results for reactions of glucosyl imidate (**7**) with DHE (**6**) under various conditions are shown in Table 1. As expected, β-anomers were primarily synthesized in the four acid catalytic conditions (Entry 1–4), and α-anomers were obtained at a 10% yield under the SiO_2_-H_2_SO_4_ acid condition (Entry 2).

The reaction was monitored by thin layer chromatography (TLC) and we confirmed that both α- and β-anomers were produced during the early stages of the reaction (the major product was β-anomers), and that α-anomers disappeared as reaction time elapsed. Therefore, an SiO_2_-H_2_SO_4_-catalyzed glycosylation reaction was done to obtain the benzoyl group-protected α-anomers, α-DHE-Glc (**4**) and α-DHE-Gal (**5**). The yields of the two α-anomers were 14% and 8.5%, respectively. OH-protected glucosyl DHE (**9** and **11**) and OH-protected galactosyl DHE (**10** and **12**) were hydrolyzed using NaOMe/MeOH to yield α- and β-anomers of DHE-Glc (**2** and **4**) and DHE-Gal (**3** and **5**), respectively (Scheme 2).

### 2.2. Effect of α- and β-Anomers of 5,6-Dihydroergosterol(DHE)-Glycosides on Cell Viability

The viability of HaCaT cells treated with α- and β-anomers was determined with MTT assays to determine experimental DHE-Glc and DHE-Gal concentrations. We have previously confirmed the effects of β-anomer DHE-sugar derivatives on cell viability [15]. As shown in Figure 2, none of the tested α- and β-anomers of DHE-Glc and DHE-Gal had an effect on cell viability up to a concentration of 20 μM. In addition, we confirmed that cell survival remained at about 80% when the concentration of the compounds was 20 μM, and cell viability at these concentrations was similar to that obtained with a 0.1% DMSO working solution alone. Therefore, all experiments were performed using concentrations up to 20 μM, and results can be interpreted independently of cell survival.

### 2.3. Effects of DHE-Glycoside α- and β-Anomers on mRNA Expression Levels of Pro-Inflammatory Cytokines and Chemokines in TNF-α/IFN-γ Induced HaCaT Cells

We examined the effects of the α- and β-anomers of DHE-Glc and DHE-Gal on pro-inflammatory cytokine and chemokine gene expression in HaCaT cells. We evaluated the mRNA expression levels of pro-inflammatory cytokines TNF-α, IL-1β, and IL-6 using an RT-PCR assay in TNF-α/IFN-γ-treated HaCaT cells. In addition, we also measured mRNA expression levels of chemokines CCL17 and CCL22, which are known to have a role in chronic skin inflammation. We confirmed that mRNA expression levels of the pro-inflammatory cytokines TNF-α, IL-6, and IL-1β were dramatically increased in TNF-α/IFN-γ-treated HaCaT cells. CCL17 and CCL22 mRNA expression levels had a similar expression pattern to TNF-α, IL-6, and IL-1β. Neither DHE (**6**), nor the DHE-glycoside α-anomers (**3** and **5**), had an effect on TNF-α, IL-6, or IL-1β mRNA levels in TNF-α/IFN-γ-treated HaCaT cells. In contrast, we found that DHE-glycoside β-anomers (**2** and **3**) reduced pro-inflammatory cytokine and chemokine mRNA expression in activated HaCaT cells (Figure 3). These results indicate that the β-anomers of DHE-glycosides were more effective than the α-anomers of DHE-glycosides at inhibiting gene expression of pro-inflammatory cytokines including IL1-β, IL-6, TNF-α and chemokines CCL17 and CCL22 in TNF-α/IFN-γ-treated HaCaT cells.

### 2.4. Effects of DHE-Glycoside α- and β-Anomers on Pro-Inflammatory Cytokine and Chemokine Protein Expression Levels in TNF-α/IFN-γ-Induced HaCaT Cells

Next, we used ELISA assays to examine whether DHE-glycoside α- and β-anomers have an effect on pro-inflammatory cytokine and chemokine protein expression in TNF-α/IFN-γ–induced HaCaT cells. In TNF-α/IFN-γ–induced HaCaT cells, TNF-α, IL-6, IL-1β, CCL17, and CCL22 protein expression increased by approximately 4- to 9-fold compared to controls. DHE treatment had no effect on pro-inflammatory cytokine protein expression in HaCaT cells relative to control cells. Moreover, TNF-α protein expression was slightly reduced compared to the positive control. HaCaT cells treated with α-anomers of DHE-Glc and DHE-Gal (4 and 5) had similar levels of IL-1β, IL-6, TNF-α, CCL17, and CCL22 protein expression as DHE-treated cells. These results were consistent with the mRNA gene expression results. However, TNF-α/IFN-γ-induced HaCaT cells treated with β-anomers of DHE-Glc and DHE-Gal (2 and 3) showed reduced pro-inflammatory cytokine and chemokine protein expression. We confirmed that the β-anomer of DHE-Glc was more effective than the β-anomer of DHE-Gal at reducing cytokine and chemokine protein expression (Figure 4). Our results indicate that the β-anomer of DHE-Glc (2) inhibited TNF-α/IFN-γ-activated expression of IL-1β, IL-6, TNF-α, CCL17, and CCL22 proteins, which resulted in the reduction of cytokine and chemokine mRNA levels in HaCaT cells.

## 3. Discussion

The genesis of this study was evaluating the anti-inflammatory activity of extract of the leaves of *Stewartia koreana* (SKE), which is a natural herb native to Korea. We found that SKE had potent anti-inflammatory activity. SKE inhibited the LPS-induced NF-κB signaling mechanism in RAW264.7 mouse macrophages, and strongly inhibited the production of NO and PGE_2_ [20]. Next, we investigated the active ingredient of SKE, and identified spinasterol-Glc (**1**) as the main active ingredient of SKE extract. Spinasterol-Glc (**1**) inhibited the production of LPS-induced NO in RAW264.7 mouse macrophage cells, in addition to inhibiting the expression of the pro-inflammatory cytokines TNF-α, IL-6, and IL-1β. This inhibition proved to be the result of inhibition of the IκB-α/IKK phosphorylation process [21]. We also found that spinasterol-Glc (**1**) strongly inhibited the expression of TARC/CCL17 stimulated by TNF-α/IFN-γ in HaCaT cells. This phenomenon was caused by suppression of phosphorylation of c-Raf, p38 MAPK, and JAK2. Furthermore, spinasterol-Glc (**1**) inhibited NF-κB and STAT1 promoter activation, and we confirmed that spinasterol-Glc (**1**) was a potential treatment option for chronic skin inflammation due to its ability to suppress CCL17 expression [3]. However, spinasterol-Glc (**1**) is difficult to isolate from natural extracts or synthesize. Therefore, we designed and synthesized an analogue of spinasterol, 5,6-dihydroergosterol (DHE). DHE has the same steroidal backbone as spinasterol, but different side chains. We have examined the biological activity of four DHE-glycosides, including ergosterol, in previous studies. We showed that DHE-glycosides strongly inhibited the production of NO induced by LPS in RAW264.7 mouse macrophages [6]. Next, we investigated the inhibitory effect of DHE-Glc on chronic skin inflammation in DNCB-induced animal models. We found that DHE-Glc inhibited the infiltration of epidermal eosinophil and mast cells in our DNCB-induced skin inflammation animal model and that DHE-Glc reduced the concentration of IgE and histamine and mRNA expression and protein levels of CCL17/22 in the plasma of DNCB-treated animals. We also confirmed that inhibition of CCL17/22 expression was due to suppression of NF-κB and STAT1 signaling [7]. Therefore, we reasoned that DHE-Glc (**2**) would be useful as a therapeutic agent for chronic skin inflammation.

In our synthesis process, however, the α- and β-anomers of DHE-glycosides have always been obtained together, particularly during the glycosylation step. To determine the relative biological activity of α- and β-anomers, we prepared α- and β-anomers of DHE-glycosides and evaluated their anti-inflammatory activity. In more detail, we synthesized α- and β-anomers of DHE-Glc and DHE-Gal (**2**, **3**, **4**, and **5**) via acid-catalyzed glycosylation of DHE (**6**) with OH-protected glucosyl trichloroimidate (**7**) or OH-protected galactosyl trichloroimidate (**8**), followed by deprotection (Scheme 2). Based on the results of glycosylation reactions in various acids (Table 1), β-anomers were synthesized by Cu(OTf)_2_-catalyzed glycosylation followed by deprotection, and α-anomers were obtained by SiO_2_-H_2_SO_4_-catalyzed glycosylation followed by deprotection. We determined the anti-chronic inflammatory activities of these compounds by measuring mRNA and protein expression levels of pro-inflammatory cytokines TNF-α, IL-6, and IL-1β and chemokines CCL17 and CCL22 in TNF-α/IFN-γ-stimulated HaCaT cells after treatment of these cells with these compounds. We found that the β-anomers of DHE glycosides had a greater inhibitory effect than the α-anomers of DHE-glycosides on the gene expression of the pro-inflammatory cytokines IL1-β, IL-6, TNF-α and the chemokines CCL17 and CCL22 in HaCaT cells. In addition, the β-anomers of DHE-glycosides decreased protein expression levels of these pro-inflammatory cytokines and chemokines. This inhibitory effect is likely due to inhibition of the phosphorylation of IKK/IκBα and activation of NF-κB and STAT1 by the β-anomers of DHE-glycosides [7]. In chronic skin inflammatory sites, keratinocytes and various immune cells are activated by inflammatory stimuli, such as TNF-α and IFN-γ [22,23]. The secretion of chemokines and pro-inflammatory cytokines from keratinocytes is responsible for chronic skin disorder diseases as well as abnormal immune responses [24,25,26]. TNF-α and IFN-γ induce strong inflammation in vivo [26,27]. According to previous reports, TNF-α and IFN-γ increased the expression and secretion of pro-inflammatory cytokines IL-1β and IL-6 and chemokines CCL17 and CCL22 in chronic skin inflammatory lesions [10,28,29,30]. Therefore, inhibiting the secretion of pro-inflammatory cytokines and chemokines is important for developing drug candidates that can effectively treat chronic skin inflammation.

We found that the β-anomers of DHE-glycosides had a greater anti-inflammatory effect than the α-anomers of DHE-glycosides. We hypothesized that the higher physiological activity of β-anomers might be explained by their greater stability in solution than the α-anomers, because some studies have reported better stability of β-anomeric glycosides than α-anomeric glycosides [31,32]. Consistent with our hypothesis, we found that the purified α-anomers of DHE-glycosides were relatively unstable in solution or in weakly acidic conditions, like silica gel, based on both thin layer chromatography (TLC) and NMR experiments. TLC experiments showed that α-anomers were not completely stable in silica gel plates, and as a result, α-anomers decomposed during silica gel chromatography purification to give low yields. In the NMR experiment, α- and β-anomers of DHE-Gal (**5** and **3**) were dissolved in DMSO-d^6^ and kept for 12 h at 60 °C, after which NMR spectra were obtained. The spectrum of DHE-β-Gal (**3**) shows that it was stable, while the spectrum of DHE-α-Gal (**5**) revealed a change in the NMR peaks from 4.5 to 5.5 ppm (Figure 5). These results confirmed that DHE-α-Gal (**5**) was unstable and that its structure changed. Based on these results, we suggest that the anti-inflammatory activity of β-anomers of DHE-glycosides is higher than that of the α-anomers because of the poor stability of the α-anomers of DHE-glycosides.

In conclusion, we designed and synthesized α- and β-anomers (**2**, **3**, **4,** and **5**) of 5,6-dihydroergosterol (DHE)-glycosides as anti-inflammatory agents and evaluated their relative anti-inflammatory activity. β-Anomers of DHE-Glc (**2**) and DHE-Gal (**3**) were prepared by Cu(OTf)_2_-catalyzed glycosylation followed by deprotection, while α-anomers of DHE-Glc (**4**) and DHE-Gal (**5**) were obtained by SiO_2_-H_2_SO_4_-catalyzed glycosylation followed by deprotection. We found that β-anomers of DHE-glycosides (**2** and **3**) reduced pro-inflammatory cytokine and chemokine mRNA expression in TNF-α/IFN-γ-activated HaCaT cells, and that the β-anomer of DHE-Glc (**2**) inhibited TNF-α/IFN-γ-activated expression of IL-1β, IL-6, TNF-α, CCL17, and CCL22 proteins, which resulted in the reduction of cytokine and chemokine mRNA levels in HaCaT cells. These results indicate that the β-anomer of DHE-Glc (**2**) had an inhibitory effect on inflammatory mediator expression in TNF-α/IFN-γ-stimulated HaCaT cells. Based on these results, we concluded that the β-anomers of DHE-Glc (**2**) and DHE-Gal (**3**) exhibited potent anti-inflammatory effects. Furthermore, the β-anomers of DHE-glycosides had a greater anti-inflammatory effect than the α-anomers of DHE-glycosides. We found that the purified α-anomers of DHE-glycosides were unstable in solution or in slightly acidic conditions based on both TLC and NMR experiments. This could explain why the anti-inflammatory activities of α-anomers of DHE-glycosides were lower than those of the β-anomers of DHE-glycosides. Further analyses are needed to determine the chemical stability of the α-anomers of DHE-glycosides. The structure-activity relationship (SAR) of various DHE-glycosides will be discussed in detail at a later date.

## 4. Materials and Methods

### 4.1. Chemicals and Instruments

^1^H NMR and ^13^C NMR spectra were recorded on a Jeol 300 MHz spectrometer and Jeol 75 MHz spectrometer (Peabody, MA, USA). Chemical shifts (δ) are reported in parts-per-million (ppm); coupling constants (*J*) are reported in Hertz (Hz). The following abbreviations were used to explain multiplicities: *s* = singlet, *d* = doublet, *t* = triplet, *q* = quartet, *m* = multiplet, and *br s* = broad singlet. Melting points were determined on a Barnstead Electrothermal 9100 instrument. HR-EI+-MS was recorded on a Jeol JMS-700 instrument. All chemicals and reagents were purchased from Sigma-Aldrich Co. Ltd. (St. Louis, MO, USA). All solvents used in reactions were purchased from Honeywell Burdick & Jackson^®^ (Muskegon, MI, USA). Reaction progress was monitored by thin layer chromatography (TLC, Merck kiesegel 60F254, Kenilworth, NJ, USA), and column chromatography was performed using a Merck silica gel 60 (230–400 mesh). 5,6-Dihydroergosterol (DHE) (**6**), 2,3,4,6-tetra-*O*-benzoyl-glucopyranosyl trichloroacetimidate (OH-protected glucosyl imidate) (**7**), and 2,3,4,6-tetra-*O*-benzoyl-galactopyranosyl trichloroacetimidate (OH-protected galactosyl imidate) (**8**) were prepared as reported previously [6,33,34].

*2,3,4,6-Tetrabenzoyl-β-glucopyranosyl dihydroergosterol* (**9**)

*2,3,4,6-Tetra-O-benzoyl-glucopyranosyl trichloroacetimidate* (**7**) (1.21 g, 1.63 mmol), 5,6-dihydroergosterol (**8**) (0.50 g, 1.25 mmol), and a 4Å molecular sieve in methylene chloride (27.0 mL) were stirred at −20 °C for 0.5 h. After addition of Cu(OTf)_2_ (49.0 mg, 0.14 mmol), the mixture was stirred at −20 °C overnight. After neutralization by Et_3_N, the mixture was filtered through celite. The filtrate was concentrated in vacuo and purified by flash column chromatography (ethyl acetate/hexane) to produce a white solid.

Yield: 68%. m.p. 160–163 °C. ^1^H-NMR (300 MHz, CDCl_3_) δ 0.51–2.10 (m, 42H, peaks from steroidal structure), 3.51–3.66 (m, 1H), 4.14–4.20 (m, 1H), 4.52 (dd, 1H, *J* = 5.9, 12.1 Hz), 4.61 (dd, 1H, *J* = 3.3, 11.9 Hz), 4.95 (d, 1H, *J* = 7.9 Hz), 5.03–5.13 (m, 1H), 5.18 (t, 2H, *J* = 6.3 Hz), 5.50 (dd, 1H, *J* = 7.9, 9.7 Hz), 5.63 (t, 1H, *J* = 9.7 Hz), 5.90 (t, 1H, *J* = 9.7 Hz), 7.29–7.59 (m, 12H), 7.82–8.03 (m, 8H).

*(3β,22E)-Ergosta-7,22-dien-3-yl-β-d-glucopyranoside (DHE-β-Glc)* (**2**)

*2,3,4,6-Tetrabenzoyl-β-glucopyranosyl dihydroergosterol* (**9**) (0.70 g, 0.72 mmol) in a mixed solvent (CH_2_Cl_2_:MeOH = 7.0 mL:7.0 mL) was added to NaOMe (0.5 M in MeOH, 8.6 mL, 4.30 mmol). The mixture was stirred at room temperature overnight. After neutralization by Dowex Mac-3, the mixture was filtered. The filtrate was concentrated in vacuo and purified by flash column chromatography (CH_2_Cl_2_/MeOH) to produce a white solid.

Yield: 89%. m.p. 267–270 °C. ^1^H-NMR (300 MHz, Pyridine-d^5^) δ 0.59 (s, 3H), 0.74 (s, 3H), 0.89 (d, 6H, *J* = 6.8 Hz), 0.99 (d, 4H, *J* = 6.8 Hz), 1.08 (d, 3H, *J* = 6.4 Hz), 1.15–1.32 (m, 4H), 1.33–1.62 (m, 8H), 1.64–1.76 (m, 4H), 1.76–1.86 (m, 2H), 1.86–1.96 (m, 2H), 1.99–2.06 (m, 3H), 3.92–4.11 (m, 3H), 4.24–4.34 (m, 2H), 4.41–4.46 (m, 1H), 4.60 (d, 1H, *J* = 11.5 Hz), 4.87 (brs, −OH peaks), 5.04 (d, 1H, *J* = 7.7 Hz), 5.19 (brs, 1H), 5.26–5.29 (m, 1H). ^13^C-NMR (75 MHz, Pyridine-d^5^) δ 12.48, 13.26, 18.05, 20.04, 20.35, 21.58, 21.96, 23.48, 28.75, 30.24, 33.56, 34.77, 34.97, 37.56, 39.85, 40.41, 41.08, 43.31, 43.72, 49.84, 55.55, 56.34, 63.19, 72.05, 75.67, 77.36, 78.83, 78.96, 80.09, 102.60, 118.23, 132.46, 136.58, 139.95. HR-EI+-MS *m*/*z*: 560.4075 (Calcd for C_34_H_56_O_6_: 560.4077).

*2,3,4,6-Tetrabenzoyl-β-galactopyranosyl dihydroergosterol* (**10**)

*2,3,4,6-Tetra-O-benzoyl-galactopyranosyl trichloroacetimidate* (**8**) (2.23 g, 3.01 mmol), 5,6-dihydroergosterol (**6**) (1.0 g, 2.51 mmol), and a 4Å molecular sieve in methylene chloride (36.0 mL) were stirred at −20 °C for 0.5 h. After addition of Cu(OTf)_2_ (90 mg, 0.25 mmol), the mixture was stirred at −20 °C for 4 h. After neutralization by Et_3_N, the mixture was filtered through celite. The filtrate was concentrated in vacuo and purified by flash column chromatography (ethyl acetate/hexane) to produce a white solid.

Yield: 74%. m.p. 158–161 °C. ^1^H-NMR (300 MHz, CDCl_3_) δ 0.50–2.35 (42H, peaks from steroidal structure), 3.56–3.75 (m, 1H), 4.27–4.38 (m, 1H), 4.40–4.47 (m, 1H), 4.66–4.72 (m, 1H), 4.93 (d, 1H, *J* = 7.7 Hz), 5.10–5.25 (m, 2H), 5.60 (dd, 1H, *J* = 3.2, 10.4 Hz), 5.75–5.81 (m, 1H), 5.99 (d, 1H, *J* = 2.8 Hz), 7.21–7.26 (m, 2H), 7.35–7.63 (m, 10H), 7.80 (d, 2H, *J* = 7.5 Hz), 7.96 (d, 2H, *J* = 7.5 Hz), 8.03 (d, 2H, *J* = 8.1 Hz), 8.11 (d, 2H, *J* = 7.3 Hz).

*(3β,22E)-Ergosta-7,22-dien-3-yl-β-d-galactopyranoside (DHE-β-Gal)* (**3**)

*2,3,4,6-Tetrabenzoyl-β-galactopyranosyl dihydroergosterol* (**10**) (0.20 g, 0.20 mmol) in a mixed solvent (CH_2_Cl_2_:MeOH = 2.0 mL:2.0 mL) was added to NaOMe (0.5 M in MeOH, 2.5 mL, 1.23 mmol). The mixture was stirred at room temperature for 5 h. After neutralization by Dowex Mac-3, the mixture was filtered. The filtrate was concentrated in vacuo and purified by flash column chromatography (CH_2_Cl_2_-MeOH) to produce a white solid.

Yield: 87%. m.p. 267–269 °C. ^1^H-NMR (300 MHz, Pyridine-d^5^) δ 0.59 (s, 3H), 0.73 (s, 3H), 0.80–0.85 (m, 1 H), 0.89 (dd, 6H, *J* = 1.9, 6.7 Hz), 0.99 (d, 4H, *J* = 6.8 Hz), 1.08 (d, 3H, *J* = 6.4 Hz), 1.18–1.62 (m, 13H), 1.68–1.72 (m, 4H), 1.77–1.86 (m, 1H), 1.88–1.94 (m, 1H), 1.98–2.17 (m, 3H), 3.92–4.07 (m, 1 H), 4.14 (t, 1H, *J* = 5.9 Hz), 4.21–4.29 (m, 1 H), 4.42–4.56 (m, 3H), 4.59–4.66 (m, 1 H), 4.97 (d, 2H, *J* = 7.5 Hz), 5.19 (brs, 1H), 5.27 (t, 1H, *J* = 5.8 Hz). ^13^C-NMR (75 MHz, Pyridine-d^5^) δ 12.46, 13.24, 18.05, 20.03, 20.35, 21.58, 21.94, 23.48, 28.74, 30.24, 30.27, 33.55, 34.73, 34.74, 34.97, 37.57, 39.85, 40.41, 41.07, 43.30, 43.70, 49.84, 55.54, 56.34, 62.85, 70.63, 72.99, 75.72, 77.24, 103.18, 118.22, 132.45, 136.57, 139.93. HR-EI+-MS *m*/*z*: 560.4075 (Calcd for C_34_H_56_O_6_: 560.4077).

*2,3,4,6-Tetrabenzoyl-β-glucopyranosyl dihydroergosterol* (**9**)

*2,3,4,6-Tetra-O-benzoyl-glucopyranosyl trichloroacetimidate* (**7**) (1.86 g, 2.51 mmol), DHE (**6**) (0.50 g, 1.25 mmol), and a 4Å molecular sieve in methylene chloride (31.0 mL) were stirred at room temperature for 0.5 h. After the addition of SiO_2_-H_2_SO_4_ (4.67 mmol/g, 27 mg, 0.125 mmol), the mixture was stirred at room temperature for 2.5 h. After neutralization by Et_3_N, the mixture was filtered. The filtrate was concentrated in vacuo and purified by flash column chromatography (ethyl acetate/hexane) to produce a white solid.

Yield: 14% (α-anomer). ^1^H-NMR (300 MHz, CDCl_3_) δ 0.50–2.06 (m, 42H, peaks from steroidal backbone), 3.26–3.40 (m, 1H), 4.05–4.13 (m, 1H), 4.35 (dd, 1H, *J* = 4.7, 12.1 Hz), 4.50 (dd, 1H, *J* = 2.8, 12.1 Hz), 4.74–4.84 (m, 1H), 5.09 (brs, 1 H), 5.17 (t, 2H, *J* = 6.1 Hz), 5.49 (d, 1H, *J* = 8.8 Hz), 5.77 (d, 1H, *J* = 1.8 Hz), 6.04 (d, *J* = 5.1 Hz, 1H), 7.19–7.29 (m, 2H), 7.36–7.52 (m, 8H), 7.55–7.66 (m, 2H), 7.75–7.83 (m, 2H), 7.88–8.00 (m, 4H), 8.05–8.13 (m, 2H).^13^C-NMR (75 MHz, CDCl3) δ 12.05, 12.88, 17.63, 19.66, 19.98, 21.12, 21.44, 22.73, 22.92, 29.74, 33.12, 34.04, 35.62, 37.16, 39.45, 40.28, 40.52, 42.87, 43.29, 49.37, 55.10, 56.00, 64.08, 67.50, 68.61, 69.25, 72.12, 73.60, 97.65, 117.52, 121.67, 126.62, 128.35, 128.40, 128.64, 128.76, 129.25, 129.35, 129.72, 129.83, 129.87, 130.11, 130.25, 132.04, 133.17, 133.70, 133.85, 135.88, 136.44, 139.52, 164.83, 165.44, 166.24 HR-FAB^+^-MS *m*/*z*: 977.5209 (M + 1) (Calcd for C_62_H_72_O_10_: 976.51).

*(3β,22E)-Ergosta-7,22-dien-3-yl-α-d-glucopyranoside (DHE-α-Glc)* (**4**)

*2,3,4,6-Tetrabenzoyl-α-glucopyranosyl dihydroergosterol* (**11**) (0.10 g, 0.10 mmol) in a mixed solvent (CH_2_Cl_2_:MeOH = 1.0 mL:1.0 mL) was added to NaOMe (0.5 M in MeOH, 1.0 mL, 0.51 mmol). The mixture was stirred at room temperature overnight. After neutralization by Dowex Mac-3, the mixture was filtered. The filtrate was concentrated in vacuo and purified by flash column chromatography (CH_2_Cl_2_/MeOH) to produce a white solid.

Yield = 28.4% (unstable). ^1^H-NMR (300 MHz, Pyridine-d^5^) 0.51–2.10 (42H, peaks from steroidal backbone), 4.30–4.53 (m, 4H), 4.59–4.71 (m, 1H), 5.00–5.20 (m, 3H), 5.22–5.31 (m, 2H), 6.49–6.60 (m, 1H). ^13^C NMR (75 MHz, Pyridine-d^5^) δ 12.30, 13.06, 17.90, 19.89, 20.21, 21.26, 21.42, 21.73, 23.32, 30.01, 33.42, 34.27, 37.38, 38.52, 40.46, 41.51, 43.16, 43.55, 49.61, 49.78, 55.37, 56.15, 62.88, 70.36, 73.18, 74.95, 76.03, 79.81, 99.49, 117.95, 121.06, 132.20, 139.75. HR-FAB^−^-MS *m*/*z*: 559.3994 (M − 1) (Calcd for C_34_H_56_O_6_: 560.4077).

*2,3,4,6-Tetrabenzoyl-α-galactopyranosyl dihydroergosterol* (**12**)

*2,3,4,6-Tetra-O-benzoyl-galactopyranosyl trichloroacetimidate* (**8**) (2.23 g, 3.01 mmol), DHE (**6**) (1.0 g, 2.51 mmol), and a 4Å molecular sieve in methylene chloride (63.0 mL) were stirred at room temperature for 0.5 h. After the addition of SiO_2_-H_2_SO_4_ (4.67 mmol/g, 54 mg, 0.25 mmol), the mixture was stirred at room temperature overnight. After neutralization by Et_3_N, the mixture was filtered through celite. The filtrate was concentrated in vacuo and purified by flash column chromatography (ethyl acetate/hexane) to produce a white solid.

Yield: 8.5% (α-anomer). ^1^H-NMR (300 MHz, CDCl_3_) δ 0.54–2.06 (42H, peaks from steroidal backbone), 3.54–3.69 (m, 1H), 4.62–4.69 (m, 1H), 4.71–4.80 (m, 2H), 5.09–5.26 (m, 3H), 5.45 (d, 2H, *J* = 12.1 Hz), 5.61 (d, 1H, *J* = 5.3 Hz), 5.97–6.10 (m, 1H), 7.25–7.33 (m, 4H), 7.34–7.47 (m, 4H), 7.48–7.61 (m, 4H), 7.92 (d, 2H, *J* = 8.3 Hz), 7.99 (d, 2H, *J* = 8.3 Hz), 8.07 (t, 4H, *J* = 8.2 Hz). ^13^C-NMR (75 MHz, CDCl3) δ 12.09, 12.99, 17.62, 19.66, 19.99, 21.14, 21.48, 22.94, 28.16, 29.40, 29.70, 33.13, 33.97, 34.35, 37.05, 39.48, 40.14, 40.57, 42.85, 43.34, 49.38, 55.15, 56.00, 63.44, 70.27, 81.97, 82.76, 104.02, 117.59, 128.55, 128.59, 129.22, 129.33, 129.68, 129.77, 129.93, 130.02, 130.16, 132.05, 133.30, 133.42, 133.52, 133.63, 135.88, 139.70, 165.77, 165.97, 166.33. HR-FAB^+^-MS *m*/*z*: 977.5209 (M + 1) (Calcd for C_62_H_72_O_10_: 976.51).

*(3β,22E)-Ergosta-7,22-dien-3-yl-α-d-galactopyranoside (DHE-α-Gal)* (**5**)

*2,3,4,6-Tetrabenzoyl-α-galactopyranosyl dihydroergosterol* (**12**) (0.17 g, 0.17 mmol) in a mixed solvent (CH_2_Cl_2_:MeOH = 1.8 mL:1.8 mL) was added to NaOMe (0.5 M in MeOH, 1.05 mL, 0.51 mmol). The mixture was stirred at room temperature overnight. After neutralization by Dowex Mac-3, the mixture was filtered. The filtrate was concentrated in vacuo and purified by flash column chromatography (CH_2_Cl_2_/MeOH) to produce a white solid.

Yield= 75.3% (unstable). ^1^H-NMR (300 MHz, Pyridine-d^5^) δ 0.58–2.17 (42H, peaks from steroidal backbone), 2.70–3.00 (m, 4H, −OH peaks), 3.71–3.87 (m, 1H), 4.39–4.47 (m, 2H), 4.55–4.65 (m, 1H), 4.88–4.97 (m, 2H), 5.10 (dd, 1H, *J* = 4.7, 6.9 Hz,), 5.21 (brs, 1H), 5.28 (t, 2H, *J* = 6.0 Hz), 5.70–5.75 (m, 1H). ^13^C-NMR (75 MHz, Pyridine-d^5^) δ 12.47, 13.26, 18.04, 20.04, 20.35, 21.58, 21.95, 23.49, 28.75, 30.24, 33.56, 34.75, 35.01, 37.54, 39.85, 40.55, 41.08, 43.31, 43.72, 46.13, 49.85, 55.55, 56.34, 65.12, 73.16, 76.66, 79.08, 84.21, 107.94, 118.21, 132.46, 136.59, 139.99. HR-FAB^+^-MS *m*/*z*: 561.4161 (M + 1) (Calcd for C_34_H_56_O_6_: 560.4077).

### 4.2. Cell Culture

Human keratinocytes (HaCaT) were cultured in Dulbecco’s modified Eagles medium (Welgene, Seoul, Korea), containing 10% fetal bovine serum (FBS), 100 units/mL penicillin, and 100 μg/mL streptomycin (Gibco BRL, Rockville, MD, USA). Cell culture conditions were as follows: 95% humidity and 5% (*v*/*v*) mixture of air and CO_2,_ in a cell culture incubator at 37 °C.

### 4.3. Cell Viability Assay

Cultured cells were removed from cell culture plates, and HaCaT cells at a concentration of 5 × 10^4^ cells/well were re-cultured in 96-well plates for 18 h. After the growth medium was discarded, HaCaT cells were treated with the indicated concentration of each compound in serum-free medium for 24 h. After removing the culture medium, cells were treated with 100 μg/mL of 3-(4,5-dimethylthiazol-2-yl)-2,5-diphenyl-thetazolium bromide (MTT) for 1 h for formazan formation in live cells. Crystallized formazan was dissolved in 200 μL DMSO and the wavelength of each sample was analyzed by spectrophotometer using an absorbance measurement at 560 nm. Analyses were repeated three times, and results are expressed as means of three independent experiments.

### 4.4. Reverse Transcriptase-Polymerase Chain Reaction (RT-PCR) Analysis

The RNA-Bee isolation kit (Tel-Test, Friendswood, TX, USA) was used according to the manufacturer’s recommendations for isolating total RNA from HaCaT cells. cDNA was prepared using reverse-transcriptase M-MuLV (Fermentas Life Science, Pittsburgh, PA, USA). PCR amplification was performed using specific primers (Bioneer, Daejeon, Korea).

### 4.5. Enzyme-Linked Immunosorbent Assay

HaCaT cells were cultured in 6-well plates and then the growth medium was removed. Cells were treated with TNF-α/IFN-γ (10 ng/mL) in the presence or absence of each compound for 18 h. Supernatants of cultured cells were collected and analyzed using an enzyme-linked immunosorbent assay (ELISA) kit (R&D Systems, Minneapolis, MN, USA) according to the manufacturer’s instructions.

### 4.6. Statistical Analysis

Unless otherwise stated, all experiments were performed with triplicate samples and repeated at least three times. Data are presented as means ± S.D. and statistical comparisons between groups were performed using 1-way ANOVA followed by a Student’s *t*-test.

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
