# Peer review of "Comparison of the Inhibitory Activities of 5,6-Dihydroergosterol Glycoside α- and β-Anomers on Skin Inflammation"

_molecules, 2019, doi:10.3390/molecules24020371_

Reviewer 1 Report

The paper at title “Comparison of Inhibitory Activities of 5,6-3 Dihydroergosterol…” whose authors are Tae Kyun Kim et al. presents (very interesting) results a synthesis and  other results novel synthesized several alfa and beta-anomers of Dihydroergosterol (DHE)-glycoside and assessed effects on CCL17 and CCL22 expression associated with anomeric structural differences. …

I think, this could be a very useful study for other authors. The paper seems to be acceptable but, in my opinion, it requires some modifications. Additionally, several questions should be answered by the authors in detail, as many important issues are described too superficially:

1. Introductions is too short. The authors should be highlight the main goal of the work more. Introduction should be completed especially about photophysical properties of new compound (Maybe in Supplementary Materials authors should add some spectra UV and FTIR).

2. Figure 2 must be improved. The quality of the presentation is really bad.

3. Discussion is too short. The authors should significantly expand this section.

4. Conclusion, is too short. This section is just a part of the discussion, which is very poor in itself.

5. References should be completed. The authors quote only 28 items!

6. Line 196: “Therefore, it could be assumed that the anti-inflammatory activities of ..-anomers of DHE glycosides are lower than those of ..anomers of DHE-glycosides due to the low stability of ..- anomers.”. Too poor conclusions, we should see some UV or FTIR spectra or other evidences.

7. Lines: 192 – 200: this section should be significantly expanded and certainly separated from the discussion if the authors want to publish this work.

8. Materials, methods are nicely described but the discussion and conclusion must be repaired.

In conclusion, the paper seems to be acceptable but requires some revisions. The whole layout and neatness of the paper do not leave too much objections, as it is prepared very carefully. Some figures must be corrected.

Please answer all my questions and comments and attach the manuscript with marked changes.

The objections presented by me do not undermine the quality of the paper, which will support in the further publishing process, certainly after careful consideration of my comments.

Author Response

1. Introductions is too short. The authors should be highlight the main goal of the work more. Introduction should be completed especially about photophysical properties of new compound (Maybe in Supplementary Materials authors should add some spectra UV and FTIR).

Answer:

Thank you for your suggestion

Introduction parts have been reorganized and expanded to explain the background of glycosides. In addition, we were adding to the highlight the main goal of this research to introduction section in page 1 line 26 to page 2 line 48, and page 2 line 77. However, in this work, we are not interested in photophysical properties of DHE-glycosides. Next time we will do more work in this area.

2. Figure 2 must be improved. The quality of the presentation is really bad.

Answer:

Figure 2 was divided into two parts to describe synthesis work clearly. Scheme 1 shows synthetic scheme and Scheme 2 describe a synthetic procedure in a detail.

3. Discussion is too short. The authors should significantly expand this section.

Answer:

Discussion parts have been reorganized and expanded to explain the history of our compound, action mechanism of DHE-Glc b-anomer, background of glycosides and biological meaning of our research data. Please see the text.

4. Conclusion, is too short. This section is just a part of the discussion, which is very poor in itself.

Answer:

Conclusion has been modified and expanded the discussion section in page 8 line 253.

5. References should be completed. The authors quote only 28 items!

Answer:

More references were added

6. Line 196: “Therefore, it could be assumed that the anti-inflammatory activities of ..-anomers of DHE glycosides are lower than those of ..anomers of DHE-glycosides due to the low stability of ..- anomers.”. Too poor conclusions, we should see some UV or FTIR spectra or other evidences.

Answer:

Additionally, we did NMR experiment for checking stability of a- and b-anomers of DHE-Gal in DMSO solution.

7. Lines: 192 – 200: this section should be significantly expanded and certainly separated from the discussion if the authors want to publish this work.

Answer:

It has been expanded from 9 lines to 18 lines.

8. Materials, methods are nicely described but the discussion and conclusion must be repaired.

Answer:

Thank you for your suggestion!

As we described before, we according to your point of view, which we have modified and expanded the discussion and conclusion sections.

In conclusion, the paper seems to be acceptable but requires some revisions. The whole layout and neatness of the paper do not leave too much objections, as it is prepared very carefully. Some figures must be corrected.

Please answer all my questions and comments and attach the manuscript with marked changes.

The objections presented by me do not undermine the quality of the paper, which will support in the further publishing process, certainly after careful consideration of my comments.

Also, we according to your opinion, which we have re-write, modified and expanded the revised manuscript in all of parts.

Thank you very much for your peer-review

Reviewer 2 Report

The authors studied the role of a novel reagent in anti-inflammation by suppressing CCL17 and CCL22. While some question need to be answered and more evidence need to be provided before its publication.

the study revealed the reagents inhibit CCL17 and CCL22 production on molecular levels. Although some indirect evidence discussed in the Introduction part indicate the CCL17/22 may induce skin inflammation, the authors need to provide some direct evidence that the new reagent synthesized exhibit biological anti-inflammation properties. e.g. mice experiment may be needed. Otherwise, there would be no biological meaning for the drug.

the structure of the manuscript need to be re-arranged. e.g. the titles in the results part need to briefly describe each result.

Please re-write the introduction part as some details need to be made.

Author Response

Answer:

Thank you for your suggestion.

In the past few years, we have been working to develop drugs for the treatment of chronic skin inflammation, likely atopic dermatitis. We previously discovered active ingredient in natural herb extracts and synthesized its derivatives. 5,6 Dihydroergosterol glycoside (DHE-Glc) improved AD-like skin inflammatory symptoms on the backs of DNCB-induced mice, partly by suppressing production of Th2 chemokines, CCL17 and CCL22 in inflamed skin. Also, we previously revealed the biological mechanism of DHE-Glc and it was confirmed to inhibit the activation of NF-kB and STAT1 signal transductions. Therefore, the development history of DHE-Glc was added to the discussion section in page 7 line 183 as follows.

The genesis of this study was evaluating the anti-inflammatory activity of extract of the leaves of Stewartia koreana (SKE), which is a natural herb native to Korea. We found that SKE had potent anti-inflammatory activity. SKE inhibited the LPS-induced NF-kB signaling mechanism in RAW264.7 mouse macrophages, and strongly inhibited the production of NO and PGE2 [20]. Next, we investigated the active ingredient of SKE, and identified spinasterol-Glc (1) as the main active ingredient of SKE extract. Spinasterol-Glc (1) inhibited the production of LPS-induced NO in RAW264.7 mouse macrophage cells, in addition to inhibiting the expression of the pro-inflammatory cytokines TNF-a, IL-6, and IL-1b. This inhibition proved to be the result of inhibition of the IkB-a/IKK phosphorylation process [21]. We also found that spinasterol-Glc (1) strongly inhibited the expression of TARC/ CCL17 stimulated by TNF-a/IFN-g in HaCaT cells. This phenomenon was caused by suppression of phosphorylation of c-Raf, p38 MAPK, and JAK2. Furthermore, spinasterol-Glc (1) inhibited NF-kB and STAT1 promoter activation, and we confirmed that spinasterol-Glc (1) was a potential treatment option for chronic skin inflammation due to its ability to suppress CCL17 expression [3]. However, spinasterol-Glc (1) is difficult to isolate from natural extracts or synthesize. Therefore, we designed and synthesized an analogue of spinasterol, 5,6-dihydroergosterol (DHE). DHE has the same steroidal backbone as spinasterol, but different side chains. We have examined the biological activity of four DHE-glycosides, including ergosterol, in previous studies. We showed that DHE-glycosides strongly inhibited the production of NO induced by LPS in RAW264.7 mouse macrophages [6]. Next, we investigated the inhibitory effect of DHE-Glc on chronic skin inflammation in DNCB-induced animal models. We found that DHE-Glc inhibited the infiltration of epidermal eosinophil and mast cells in our DNCB-induced skin inflammation animal model and that DHE-Glc reduced the concentration of IgE and histamine and mRNA expression and protein levels of CCL17/22 in the plasma of DNCB-treated animals. We also confirmed that inhibition of CCL17/22 expression was due to suppression of NF-kB and STAT1 signaling [7]. Therefore, we reasoned that DHE-Glc (2) would be useful as a therapeutic agent for chronic skin inflammation.

Also, we according to your opinion, which we have re-write, modified and expanded the revised manuscript in all of parts.

Thank you very much for your peer-review

Round  2

Reviewer 2 Report

good to be published